# Tetranuclear Copper(I) and Silver(I) Pyrazolate Adducts with 1,1′-Dimethyl-2,2’-bibenzimidazole: Influence of Structure on Photophysics

**DOI:** 10.3390/molecules28031189

**Published:** 2023-01-25

**Authors:** Gleb B. Yakovlev, Aleksei A. Titov, Alexander F. Smol’yakov, Andrey Yu. Chernyadyev, Oleg A. Filippov, Elena S. Shubina

**Affiliations:** 1A. N. Nesmeyanov Institute of Organoelement Compounds, Russian Academy of Sciences, Vavilov Str., 28, 119334 Moscow, Russia; 2A. N. Frumkin Institute of Physical Chemistry and Electrochemistry, Russian Academy of Sciences, Leninsky prosp. 31/4, 199071 Moscow, Russia

**Keywords:** cyclic trinuclear complex, pyrazolate adducts, copper(I), silver(I), TD-DFT, photoluminescence

## Abstract

A reaction of a cyclic trinuclear copper(I) or silver(I) pyrazolate complex ([MPz]_3_, M = Cu, Ag) with 1,1′-dimethyl-2,2’-bibenzimidazole (**L**) leads to the formation of tetranuclear adducts decorated by one or two molecules of a diimine ligand, depending on the amount of the ligand added (0.75 or 1.5 equivalents). The coordination of two **L** molecules stabilizes the formation of a practically idealized tetrahedral four-metal core in the case of a copper-containing complex and a distorted tetrahedron in the case of a Ag analog. In contrast, complexes containing one molecule of diimine possess two types of metals, two- and three-coordinated, forming the significantly distorted central M_4_ cores. The diimine ligands are twisted in these complexes with dihedral angles of ca. 50–60°. A TD-DFT analysis demonstrated the preference of a triplet state for the twisted 1,1′-dimethyl-2,2’-bibenzimidazole and a singlet state for the planar geometry. All obtained complexes demonstrated, in a solution, the blue fluorescence of the ligand-centered (LC) nature typical for free diimine. In contrast, a temperature decrease to 77 K stabilized the structure close to that observed in the solid state and activated the triplet states, leading to green phosphorescence at ca. 500 nm. The silver-containing complex Ag_4_Pz_4_L exhibited dual emission from both the singlet and triplet states, even at room temperature.

## 1. Introduction

Coinage metal complexes have generated considerable interest in the last few decades because of their rich coordination chemistry and the properties determining their different practical uses in areas such as catalysis, the development of light-emitting devices [1], antibacterial/antimicrobial agents [2], etc. The fashioning of new expanded multinuclear metal complexes is an important step toward versatile multidimensional compounds and materials with specified structures and given properties. The employment of bridging ligands allows the construction of such systems. For example, bridging ligands are very popular for the synthesis of metal–organic frameworks (MOF) [3,4,5]. Cooper(I) and silver(I) complexes are of constant interest due to their broad and interesting photophysical properties [6,7,8,9,10,11,12,13].

The pyrazolate anion is a bridging bidentate ligand that can also play the role of a counterion [14,15,16,17]. Copper(I) and silver(I) pyrazolate adducts, which often possess isostructural complexes, have a special place in coordination chemistry. Depending on the reaction conditions and substituents in the initial pyrazole, poly- [18], di- [19,20], tri- [21,22], and tetranuclear [23,24] complexes can be obtained. Cyclic trinuclear complexes (Figure 1, [MPz]_3_) are more popular because of their planar structures, allowing the formation of supramolecular aggregates via M-M or M…π interactions [15,22,25]. Being polynuclear Lewis acids, cyclic copper(I) and silver(I) pyrazolates coordinate bases that possess π-electron density, such as aromatic compounds, alkenes, and alkynes [26,27,28,29,30,31,32]. However, bases of other types also form stable complexes with [MPz]_3_ [33,34,35,36,37,38]. Thus, the interactions of cyclic trinuclear complexes with nitrogen- or phosphorus-containing ligands lead to the formation of heteroleptic metal pyrazolates of different structures, depending on the reagent ratios [39,40,41,42,43,44]. The complexation of macrocycles with diimines (bipyridine- or phenanthroline-type ligands) allows di-, tetra-, or pentanuclear complexes to be obtained [40,45,46].

Modifying pyrazoles with the introduction of pyridine yields chelating ligands. In this case, mixed-ligand mononuclear metal complexes with chelating or bridging bisphosphines have been obtained [47,48].

Another example of a diimine-type ligand is 2,2’-bibenzimidazole and its derivatives. In the NH form, it could serve as a chelating ligand [49,50,51]. The deprotonation of the NH group following the interaction with metal salts allows neutral chain-type complexes to be obtained. The presence of substituents (Me, for example) at these nitrogen atoms leads to steric repulsions and forces the ligand to serve as a bridge [52]. 

In this paper, we report the syntheses and structures of a series of copper(I) and silver(I) 3,5-bis(trifluoromethyl)pyrazolate (3,5-(CF_3_)_2_Pz, Pz) adducts with 1,1′-dimethyl-2,2’-bibenzimidazole (**L**) (Figure 1). An excess of **L** led to the formation of tetranuclear complexes containing two molecules of a bibenzimidazole derivative, whereas a deficiency of **L** led to the formation of a similar tetranuclear core containing only one ligand molecule. All obtained complexes exhibited ligand-centered fluorescence in solution and dual-emissive behavior in the solid state. 

## 2. Results and Discussions

### 2.1. Synthesis

1,1′-Dimethyl-2,2’-bibenzimidazole was synthesized by the aerobic oxidative homocoupling of 1-methylbenzimidazole according to the published procedure [53]. Macrocycles of [CuPz]_3_ and [AgPz]_3_ were obtained by refluxing the mixture of the corresponding metal(I) oxide and 3,5-bis(trifluoromethyl)pyrazole in toluene overnight [21]. 

According to previous results on the interaction of trinuclear Cu(I) and Ag(I) complexes with diimines [40,45,46], the reaction outcome depends on the macrocycle/ligand ratio. Therefore, we tested three possibilities for 1,1′-dimethyl-2,2’-bibenzimidazole, taking 0.75, 1, and 1.5 equivalents of **L** relative to the macrocycle. The interactions with 0.75 and 1.5 equivalents of **L** gave individual compounds that were isolated by crystallization. In the case of an equimolar amount of **L,** crystallization also gave white solids, but further analysis revealed they were mixtures of complex **1** or **3** containing two ligands (similar to those obtained with 1.5 equivalents of L) and an unreacted macrocycle. This suggests the preferential formation of a more stable and idealized structure with two molecules of 1,1′-dimethyl-2,2’-bibenzimidazole.

Compounds **1**–**4** were air-stable in the solid state and soluble in chlorinated low-polar and polar organic solvents but insoluble in water, diethyl ether, and alkanes. Compounds **1**, **3**, and **4** were stable in solution, even under air. Only the dichloromethane solution of complex **2** in the presence of moisture formed a non-significant amount of blue precipitate after staying in air for 24 h. Nevertheless, the subsequent decantation of colorless solution followed by precipitation with hexane gave pure complex **2**.

The dynamic behavior was present in the solutions of all complexes, corresponding with the broadening signals in the ^1^H NMR spectra. Complexes **2**–**4** demonstrated the typical signals of pyrazolate fragments, and L non-significantly shifted during the complexation. In the case of complex 1, possessing the most symmetrical geometries, two sets of Me^L^ signals in the ^1^H spectrum and CF_3_^Pz^ signals in the ^19^F spectrum demonstrated the overall retention of the complex geometry in solution on the NMR time scale (see the Appendix A).

### 2.2. Crystal Structures

The crystals of complexes **1**–**4** were obtained by the slow evaporation of a CH_2_Cl_2_/hexane (*v*/*v* = 1/2) solution at 5°. Single-crystal X-ray diffraction analyses revealed that all complexes were tetranuclear metal complexes. It should be noted that only the results of the XRD analyses could shed a light on the compositions of the complexes because there is periodic dependence in the case of cyclic compounds, and different spectroscopic methods and elemental analyses could only demonstrate the ratio of pyrazolate and diimine ligands. Obtaining crystals for X-ray diffraction is an essential part of such works. Despite several attempts to obtain good-quality crystals, in the case of complexes **1**–**3**, they were twins.

Complexes **1** and **3** are tetranuclear pyrazolate adducts containing two molecules of L (Figure 1). The molecule of **1** in the crystal is in a special position and possesses a high level of symmetry (contains a four-fold rotoinversion axis). That leads to the practically idealized location of copper atoms in the vertex of the tetrahedron containing four edges of 3.333(2) Å and two of 3.369(2) Å. In contrast, metal atoms in the silver-containing analog form significantly distorted tetrahedrons with edge lengths in the range 3.278(2)–3.610(4) Å. Such central M_4_N_8_ cores are typical for free tetranuclear coinage-metal pyrazolate adducts, but the coordination of additional ligands leads to the elongation of central M-M distances and increases in MNN angles [22]. Moreover, complexation leads to a significant elongation of the M–N^Pz^ bond length of ca. 0.1 Å in comparison with free trinuclear complexes of [MPz]_3_ [22]. Interestingly, metal atoms in copper- and silver-containing complexes possess similar M-N bond lengths with pyrazolate and bisdiimine ligands. For example, the Cu-N^Pz^ bond lengths in **1** are 1.98(2) and 1.99(2) Å, and both Cu-N^L^ bonds are equal 1.97(2) Å. In the case of complex **2**, the corresponding bonds are in the ranges of 2.19(2)–2.25(2) and 2.20(2)–2.29(2) Å.

All metal atoms in complexes **1** and **3** are three-coordinated by nitrogen atoms from two pyrazolate (N^Pz^) ligands and one 1-methylbenzimidazole (N^L^) core. Copper atoms in **1** are non-significantly exposed out of the N^Pz^N^Pz^N^L^ planes by 0.086 Å. In the case of the less-symmetrical complex **3,** this displacement is bigger, in the range of 0.094–0.202 Å. The 2,2’-bibenzimidazole moieties are significantly twisted in both complexes via the C-C bond, adopting the anti configuration of unsubstituted nitrogen atoms. The angles between the planes of the corresponding benzimidazole cores are 61.29° in complex **1** and 51.73 and 57.43 in complex **3**. Both complexes are additionally stabilized by intramolecular π^Pz^–π^L^ stacking interactions formed by each pyrazolate ligand and the bisimidazole fragment of L. The shortest π^Pz^–π^L^ distances are 3.213 Å and 3.270 Å for **1** and **3**, respectively.

Complexes **2** and **4** contain one molecule of diimine ligand that leads to the distorted M_4_N_8_ central core (Figure 2). Two metal atoms that participate in the interaction with **L** are three-coordinated, and two other metal atoms are two-coordinated with a linear arrangement of corresponding pyrazolate nitrogens. These two types of metal coordination spheres differ in M–N^Pz^ bond lengths and MNN angle values. The metals coordinated by pyrazolate ligands only feature bond lengths similar to those in free [MPz]_3_ macrocycles: av. M–N^Pz^ is 1.9 Å for copper-containing compounds and 2.1 Å for silver-containing compounds. In contrast, the geometry parameters for three-coordinated metals are similar to those in complexes **1** and **3** (elongation of M–N^Pz^ at ca. 0.1 Å). These compounds are also stabilized by intramolecular π^Pz^–π^L^ stacking interactions. The presence of shortened intramolecular Ag-Ag contacts (3.205(1)–3.211(1) Å) is established in complex **4**.

In a crystal, all complexes realize supramolecular packing via the network of CH…π, CH…F, and F–F intermolecular interactions.

### 2.3. Photophysical Properties

The UV-vis absorption spectra of diluted dichloromethane (DCM) solutions of complexes **1**–**4** (c = 1 × 10^−5^ M) demonstrated slightly structured bands centered at ca. 300 nm (Figure 3). All observed absorptions were identical to each other and to the absorption of the free ligand **L**. This means that in all cases the electronic transitions of the π→π* character within the 1,1′-dimethyl-2,2’-bibenzimidazole were observed. The room temperature (RT, 298 K) emission spectra of these diluted solutions of **1**–**4** (Figure 3) displayed a structured band centered at 370 nm, indicating fluorescence of an ligand centered (LC) origin. The intensities of these bands did not depend on the presence of air, demonstrating the emission from an excited singlet state.

Such spectral behavior demonstrates the possibility of complex dissociation at this concentration. Previously, it was shown that the composition of trinuclear silver(I) pyrazolate complexes depends on the concentration [54,55]. The formation of aggregated complexes (trinuclear) is preferred at a concentration higher than 7 × 10^−3^ M, but at lower concentrations (up to ~5 × 10^−4^ M), dissociated metal–pyrazolate adducts form (mononuclear salts). The investigation of the photophysical properties of **1**–**4** was performed at high concentrations (c = 1.2 × 10^−2^ M), leading to aggregated complexes in the thin cuvette for UV-vis measurements (d = 0.047 mm) and thin quarts capillaries (d~0.5 mm) for the photoluminescence. Significant spectral changes were observed. The UV-vis absorption spectra of concentrated solutions of complexes **1**–**4** in dichloromethane (c = 1.2 × 10^−2^ M) demonstrated structureless broad high-energy bands at ca. 290 nm and three bands similar to those observed in the diluted solution at 314, 330, and 345 nm (Figure 4, left). The RT emission spectra of **1**–**2** demonstrated bands at 356, 370, 388, 410, and 440 nm. Different contributions of the band intensities were observed (Figure 4, right). In the case of complex **2**, the intensities of the two central bands were comparable. Complexes **3**–**4** displayed a practical absence of the high-energy band at 356 nm and three bands similar bands to those in **1**. Such differences in the spectral behaviors of the diluted and concentrated solutions demonstrated the role of complexation. It is evident that at low concentrations the tetranuclear core dissociates. But, the presence of a low-energy phosphorescent band in photoluminescent spectra of diluted solutions, especially in the case of silver–containing complexes, demonstrates the existence of mono- or dinuclear metal–benzimidazole complexes containing pyrazolate anions as counterions.

The temperature decrease to 77 K resulted in splitting the emissions of concentrated and diluted solutions into two components (Figure 5). Copper-containing complexes **1** and **2** at low concentrations displayed a low, intense, broad band centered at ca. 500 nm. In contrast, silver-containing analogs demonstrated a slightly structured band centered at ca. 480 nm. The intensities of high-energy bands in these cases were higher. A concentrated solution of complex **2** practically did not show the high-energy band. In contrast, complexes **3** and **4** showed a non-intense structured high-energy band. The positions of these bands were closer to those observed for diluted frozen solutions of all complexes and similar to the observed RT spectra. The frozen solution of concentrated complex **1** exhibited a structured high-energy band with a maximum at 400 nm, demonstrating the different emission behavior. Copper-containing complexes **1** and **3** displayed unstructured broad bands with larger Stokes shifts, indicating the ^3^MLCT nature of the green emission. In contrast, silver-containing complexes **2** and **4** possessed structured low-energy bands. This indicated the ^3^LC emission within the diimine ligand, which was due to the effect of a heavy silver atom (spin-orbit coupling, SOC).

In the solid state, under excitation by UV irradiation (305 nm), the emissions of complexes **2**–**4** were similar to those in the concentrated solutions (Figure 6). Complex **2** displayed an unstructured broad band at 360 nm with a shoulder at 410 nm and a non-intense tail centered at ca. 520 nm. Complex **3,** with two molecules of **L,** demonstrated a structured band at 370 nm, typical for ligand-centered fluorescence, and a low-energy non-intense shoulder. Complex **4** exhibited a narrow high-energy band with a maximum at 370 nm possessing less structure and a broad band at 490 nm. In contrast, complex **1** demonstrated two unstructured bands at 380 and 426 nm and a low, intense, structured tail at 525 nm, similar to that observed for **2**. As in the solutions, the temperature decrease to 77 K led to the emission splitting into two components: fluorescence bands in the higher-energy region and phosphorescence in the lower-energy region (Table 1).

We measured the phosphorescence spectra (Figure 7) and established the lifetime (0.523 ms) of complex **3**. They demonstrated a good match with a band observed at 490 nm in a steady-state experiment, revealing the triplet nature of this band. The intensity of the low-energy shoulder in the case of complex **3** did not allow us to measure the phosphorescence spectrum at RT. This shoulder and the tail in the case of complex **2** were also definitely related to phosphorescence. This tail could be assigned to the ^3^LC^L^ phosphorescence. In the solid state, the nature of the high-energy bands could be related to the fluorescence of ^1^ML^L^CT. The measured phosphorescence spectra of complexes **1**, **3**, and **4** and the established lifetimes of all complexes in the microsecond domain (Table 1) prove this suggestion. It should be noted that such long-lived phosphorescence was observed for silver pyrazolate adducts with diimines [45,46]. Interestingly, in the case of complex **1**, the intensity of the phosphorescence band was negligibly small in comparison with the fluorescence band.

To describe the phenomena that observed, a TD-DFT analysis was performed. It was shown that there are two possible geometries of L in the gas phase. The *anti*-geometry is a flattened planar with the opposite orientation of the Me^L^–substituents, and the *syn*-geometry is a twisted molecule, as shown in Figure 2:

The *anti*-geometry is preferred by 2.7 kcal/mol (both on the ΔH and ΔG scales). Both conformers were calculated to have similar fluorescence energy (ca. 360–370 nm), but they differ in the emission shape, which was vibrationally resolved in the case of the *anti*-conformer. The reason for this is an S_1_ state geometry, which is completely flattened for the *anti*-conformer (NC-CN dihedrals are 144° in GS and 179° in S_1_, Figure 8), while for the *syn*-conformer this dihedral changes from 63° to 24°. Obviously, such flattening entails a decrease in the angle between the N-bound methyl groups (^N^CH_3_), which is highly unfavorable, giving this exited state too much energy (+9.2/+8.8 in ΔH/ΔG terms).

The geometries of the lowest triplet excited states (in the TD framework) are almost the same as the S_1_ ones, with the NC-CN dihedrals being 178° and 26° for the *anti* and *syn* rotamers, respectively. This finally leads to a completely different phosphorescence efficiency, with calculated phosphorescence rate constants of 4.7 and 2.2 × 10^3^ s^−1^, respectively, meaning a three-order better phosphorescence process for the *syn-*conformer. This also reflects the absence of the 1,1-dimethyl-2,2’-bibenzimidazole phosphorescence in solution, where rotations are fully unrestricted and an *anti*-conformer with a scarce triplet emission dominates. The *syn*-geometry present in the metal complexes suggests phosphorescence that should be further amplified due to the heavy atom effect.

Indeed, silver complexes **3** and **4** have the lowest-energy singlet transition orbitals, which are almost the same as those of the *syn*-**L** conformer. The energies of the S_1_ (and pairwise S_2_ in the case of **3,** with two independent **L** ligands) transitions are slightly lower than in free **L**, at 39370.7 and 39578.2 cm^−1^ (254.0 and 252.7 nm) when keeping the same high oscillator strengths (0.9648 and 0.8860). The two lowest singlets in **3** are independent transitions since the Ag atom participation in these transitions is less than 0.5% and there are literally two moles of Ag_2_**L** complex per mole of **3**. That also affects the absorption intensity, which is ca. two times higher for complex **3** compared to **4** at the same concentration.

Copper complexes **1** and **2** are different cases. For **2,** the LC transition of ligand **L** becomes S_9_ (39,090 cm^−1^, 255.8 nm, f = 0.5938) and similarly shifts to the lower energy, as in the case of silver but with half the oscillator strength. In the case of **1,** the lowest singlet state with notable involvement of the **L** ligand is an S_19_ (and **L** involvement is unexpectedly small at 27%, and the other 70% comes from Cu atoms). The lowest-energy singlets for copper complexes are MLCT transitions; the strengths of these transitions are ca. one order lower for complex **1**. The reason for the low intensity of complex **1** is the symmetry of these transitions, which involve a whole highly symmetric complex (in contrast to the silver case, wherein the complex of the symmetry of two independent transitions is realized).

To search deeper, we checked the behavior restricted to S_4_ geometry **1** and the unrestricted behavior. The symmetry constraints did not change the global pattern of electronic transitions but allowed us to find a metal–centered (MC) transition. This transition was hidden in the S_0_→S_1_ transition of **1**^S4^, which was responsible for 27% of the transition electron density, and more importantly it was the only one in which symmetry was allowed (Figure 9). The same MC natural transition orbital could be found in unrestricted complex **1**^nosymm^ at the almost same energy (314.5 vs. 313.6 nm). Complex **2** (as well as silver analogs **4** and **3**) also possesses a similar transition orbital, but it appears in the S_0_→S_6_ transition, which lies > 4600 cm^−1^ higher, which makes it ineffective for luminescence. In the case of silver, MC transitions are also ineffective since the lowest one is an LC with a very high oscillator strength.

## 3. Materials and Methods

### 3.1. Physical Measurement and Instrumentation

^1^H, ^19^F, and NMR measurements were carried out using a Bruker Avance 400 spectrometer (Bruker, Billerica, MA, United States). Infrared (IR) spectra were collected using a Shimadzu IRPrestige 21 FT-IR spectrometer (Shimadzu, Kyoto, Japan). in nujol. The UV-vis spectra of the solutions were measured using a Cary 50 (Varian). The photoluminescence spectra in the solution were measured using a Shimadzu RF-6000 (Shimadzu, Kyoto, Japan). The photoluminescence spectra of solid samples and frozen solutions, and lifetime measurements of the phosphorescence were recorded at 77 K and 298 K using a Fluorolog-3 spectrofluorometer system (HORIBA Jobin Yvon S.A.S., Palaiseau France) (the excitation source was a 450 W Xenon lamp with Czerny–Turner double monochromators; the registration channel was a R928 photomultiplier, while a 150 W pulsed Xenon lamp (HORIBA Jobin Yvon S.A.S.) was used for the lifetime measurements). The samples for these measurements were packed in quartz capillaries. The phosphorescence quenching curves were analyzed using the FluoroEssence^TM^ (HORIBA Jobin Yvon S.A.S., Palaiseau France) software for the calculation of the phosphorescence lifetime values.

### 3.2. Crystal Structure Determination

Single-crystal X-ray diffraction experiments of **1**–**4** were carried out with a Bruker SMART APEX II diffractometer (Bruker, Billerica, MA, United States). The APEX II software [56] was used to collect frames of data, index reflections, determine lattice constants, and integrate the intensities of reflections as well as for scaling and absorption correction. The structures were solved using a dual-space algorithm and refined in an anisotropic approximation for non-hydrogen atoms against F 2 (*hkl*). Hydrogen atoms of methyl, methylene, and aromatic fragments were calculated according to those idealized geometries and refined with constraints applied to C-H and N-H bond lengths and equivalent displacement parameters (U eq (H) = 1.2U eq (X), X—central atom of XH₂ group; U eq (H) = 1.5U eq (Y), Y—central atom of YH₃ group). All structures were solved with the ShelXT [57] program and refined with the ShelXL [58] program. Molecular graphics were drawn using the OLEX2 [59] program.

### 3.3. Computational Details

The DFT computations performed with the ORCA 5.03 package [60,61], applying ωB97X-D3 functional [62,63] and the SVP/TZVP [64] basis set with the ZORA relativistic Hamiltonian. The free 1,1-dimethyl-2,2’-bibenzimidazole was calculated by applying the TZVP basis set, while for the metal complexes M_4_Pz_4_L_x_ (x = 1, 2) the SVP basis set was applied. The ground states of ligand **L** and all complexes were fully optimized without any constraints. Additionally, copper complex **1** was optimized with the S_4_ geometry, reflecting the obtained crystal structure. No essential differences in the energies or characteristics of the electronic transitions between the two geometries of **1** were observed.

The RIJCOSX procedure was used to speed up calculations. The 20 lowest-energy excited states (40 for copper complexes **1** and **2**) were considered under the Tamm–Dancoff approximation (TDA). A spin-orbit coupling (SOC) matrix was obtained using the SOMF(1X) method [65]. Optimizations of the excited states were performed in the TD-DFT formalism for the first singlet and triplet for both the *syn-* and *anti-*conformers of **L**. The modeling of the absorbance, fluorescence, and phosphorescence spectra and rate constants were conducted with the ORCA excited states dynamics (ESD) module [66]. The quantitative analysis of excitations (natural transition orbitals and fragment impacts) was performed using the Multiwfn v3.7 program [67].

### 3.4. Synthesis and Characterization

All reactions were performed under an argon atmosphere using anhydrous solvents or solvents treated with an appropriate drying reagent. The diimine ligand 1,1’-dimethyl-2,2’-bis-benzimidazole [53] and [ML]_3_ [21] were synthesized as described.

General procedure for the synthesis of complexes **1**–**4**: The solution of [MPz]_3_ (0.0625 mmol, M = Cu (50 mg), M = Ag (58.3 mg)) and a corresponding amount of **L** (0.75 eq. (12.3 mg); 1.5 eq. (24.6 mg)) was stirred in toluene (2 mL) overnight at room temperature. The solvent was evaporated to dryness under reduced pressure, and the residue was dissolved in 1 mL of CH_2_Cl_2_. Hexane (2 mL) was added to the solution, and the powder (pale yellow for complex 1 and white for complexes **2**–**4**) was obtained by crystallization at −10 °C.

Complex **1**: Yield: 56.7 mg (76%). ^1^H NMR (300 MHz, CD_2_Cl_2_): δ = 3.56 (s, 6H, CH_3_^bzim^); 4.36 (s, 5.80H, CH_3_^bzim^); 6.70–6.81 (m, 3.94H, H^Pz^); 7.41–7.53 (m, 12.20H, H^Ar^); and 7.83 (br.s, 3.80H, H^Ar^). ^19^F NMR (282 MHz, CD_2_Cl_2_): δ = -61.30; -60,96 (CF_3_). IR (KBr, cm^−1^): 3155, 3068, 2922, 2851, 1260, 1122, 1015, and 740. Elemental analysis: C_52_H_32_Cu_4_F_24_N_16_. *Found*/*Calc.* (%) = **C** 39.07/39.25; **H** 2.18/2.03; and **N** 14.13/14.09.

Complex **2**: Yield: 54.8 mg (88%). ^1^H NMR (300 MHz, CDCl_3_): δ = 3.57 (s, 6H, CH_3_^bzim^); 6.69 (br.s, 9.96H, H^Pz^); 7.2 (d, 1H, H^Ar^); 7.35–7.37 (m, 1.05H, H^Ar^); 7.54–7.57 (m, 4H, H^Ar^); and 7.84–7.87 (m, 1.95H, H^Ar^). ^19^F NMR (282 MHz, CDCl_3_): δ = −61.37; −60.70 (m, CF_3_). IR (KBr, cm^−1^): 3154, 3068, 2926, 2854, 1268, 1124, 1026, and 744. Elemental analysis: C_36_H_18_Cu_4_F_24_N_12_. *Found*/*Calc.* (%) = **C** *32.48*/*32.54*; **H** *1.48*/*1.37*; and **N** *12.53*/*12.65.*

Complex **3**: Yield: 74.6 mg (90%). ^1^H NMR (300 MHz, CD_2_Cl_2_): δ = 4.05 (s, 12H, CH_3_^bzim^); 6.75 (s, 3.80H, H^Pz^); 7.41–7.50 (m, 12H, H^Ar^); and 7.73 (d, 4H, H^Ar^). ^19^F NMR (282 MHz, CD_2_Cl_2_): δ = −61.11 (s, CF_3_). IR (KBr, cm^−1^): 3145, 3067, 2926, 2855, 1261, 1128, 1009, and 741. Elemental analysis: C_52_H_32_Ag_4_F_24_N_16_. *Found/Calc. (%) =* C 35.18/35.32; H 1.93/1.82; and N 12.58/12.67.

Complex 4. Yield: 57.2 mg (81%). ^1^H NMR (300 MHz, CD_2_Cl_2_): δ = 3.76 (s, 6H, CH_3_^bzim^); 6.77 (s, 4H, H^Pz^); 7.37–7.51 (m, 6.05H, H^Ar^); and 7.706 (d, 2H, H^Ar^). ^19^F NMR (282 MHz, CD_2_Cl_2_): δ = −61.18 (s, CF_3_). IR (KBr, cm^−1^): 3158, 2963, 2919, 2852, 1261, 1127, 1016, 811, and 746. Elemental analysis: C_36_H_18_Ag_4_F_24_N_12_. *Found*/*Calc.* (%) = **C** 28.64/28.71; **H** 1.28/1.20; and **N** 11.03/11.16.

## 4. Conclusions

Four new tetranuclear copper(I) and silver(I) pyrazolate adducts were synthesized and characterized. Depending on the initial reagent ratios, complexes with one and two molecules of 1,1′-dimethyl-2,2’-bibenzimidazole could be obtained. In the case of complexes with two molecules of **L,** a practically idealized tetrahedral four-metal core was synthesized in the case of copper-containing complexes, and a distorted tetrahedron was synthesized in the case of the Ag analog. In contrast, complexes containing one molecule of diimine possessed two types of metals: two- and three-coordinated, leading to significantly distorted central M_4_ cores. All obtained complexes possessed dissociation in solution. At low concentrations, the non-aggregated complexes preserving the coordination of the metal centers to the 1,1′-dimethyl-2,2’-bibenzimidazole were preferred. The typical free diimine blue fluorescence of a ligand-centered (LC) nature with maxima at ca. 370 nm was observed for the solutions with compounds at low concentrations (1 × 10^−5^). In contrast, high concentrations led to aggregated complexes similar to those in the solid state. Complex formation allowed us to stabilize an unfavored twisted syn-conformer of 1,1′-dimethyl-2,2’-bibenzimidazole that was phosphorescent, in contrast to the planar anti-conformer. This allowed the observation of ^3^LC phosphorescence in the cases of **3** and **4,** even at 298 K. The copper coordinated to **L** led to the appearance of low-energy MLCT electronic transitions with very low intensity (oscillator strength) that resulted in a loss of LC^L^ phosphorescence intensity.

## Data Availability

The data presented in this study are available in article and Appendix A.

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
