# Peer review of "Tetranuclear Copper(I) and Silver(I) Pyrazolate Adducts with 1,1′-Dimethyl-2,2’-bibenzimidazole: Influence of Structure on Photophysics"

_molecules, 2023, doi:10.3390/molecules28031189_

Round 1

Reviewer 1 Report

The research article submitted by Aleksei A. Titov and co-authors contributes to the hot topic in the chemistry and material science of the development of photoactive transition metal complexes. This manuscript describes the preparation and characterization of the four tetranuclear copper and silver pyrazolate adducts accompanied by photophysical studies and theoretical insights. 

As NMR studies evidence the equilibria in the solution, the presented description of the photophysical behavior in the solution makes absolutely no sense to keep it in the manuscript in its current form, as no separate species were observed as well as their composition and quantity were not found. It is unlikely that the given information also is correct and was measured for all the complexes with diluted solutions as mentioned in Figure 3 (C = 2*10-6 M). Moreover, time-resolved data measured for the high-energy band is not given either for the solution or solid state at both 77 and 298 K. Authors are very welcome to explain their statement about similarities of the emission spectra in the solution and solid states, as at 298 K none of the solid (except 3) shows structured profiles. It is clear also as most of them also demonstrate shits of the maxima. A more detailed description of behavior 2 in the solid state is necessary, it is a pity that the time-gated experiment for 2 is missing in Figure 6. The nature of the transitions also could be proven by demonstrating excitation spectra at both low- and high-energy bands. The authors strongly encouraged revisiting the photophysical part.

In view of the fact that crystallographic data was not supplemented with checkcif reports, the quality of the given structures is unclear. Depositing the data files to the CCDC structural database is not an indication of the quality. The generation of the reports (from provided cif) reveals a few B alerts, which are necessary to be explained and resolved prior next round of revision.

A significant deviation of the CHN analysis is found for complex 3, any explanation for the purity of the substance? Would be great to see a PXRD confirming the phase purity. 

It is unclear what the authors meant by saying (lines 24-25) “In contrast, the temperature decrease to 77 K stabilizes the structure close to that observed in the solid state and activates the triplet states leading to green phosphorescence at ca. 500 nm.” Rigidification of the molecule leading to a suppression of the non-radiative channels?

Quite many typos were left in the manuscript, authors are also responsible for making the text more clear: line 103 – solution at 5°; line 108 – is an essential t part of such works; line 186 – (spin-orbital coupling, SOC); line 235 – phsosporescence; etcetera. 

Despite a few major issues this manuscript could be published on Molecules if the next revised version will be improved.

Author Response

Please, find attached. 

Reviewer 2 Report

In this work, authors report four interesting luminescent Cu(I) and Ag(I) complexes, obtained by treatment of famous pyrazolate Cu3/Ag3 triangles with 2,2'-bibenzimidazole. The resulting four-nuclear complexes were exhaustively studied both in solid and in solution state. Moreover, their electronic structures and the nature of the electronic transitions were calculated at DFT/TDDFT level of theory. The emission properties were also studied for a solution and solid states to reveal an interesting dual emission behaviour for solid-state emission. Overall, this work seems sound, original, and well-designed. Thus, I recommend acceptance of this manuscript after addressing the following concerns:

1. It would be desirable to insert excitation spectra recorded for both emission bands of the complexes presented. If available, please add these plots.

2. The emission lifetimes of 3 and 4 seem to be quite long at both temperatures. So, I recommend adding 1-2 sentences, in which these lifetimes are compared with those of known Ag(I)-pyrazolate complexes.

3. Is the emission of cluster 1 sensitive to excitation wavelengths? Anyway, please add the excitation wavelength used for recording emission curves on Figs. 4 and 5. 

4. Finally, I recommend to kindly mention some other works on luminescent Cu(I) and Ag(I) complexes, i.e.: DOI:10.1039/C8QI01302K, 10.1002/anie.202103037, 10.1002/chem.201402060,10.1039/C6CC06613E, 10.1021/jacs.7b04550, 10.1016/j.ccr.2022.214975.

Author Response

Please, find attached. 

Round 2

Reviewer 1 Report

Aleksei A. Titov and co-authors improved the manuscript by correcting a few major points, thus it can be published in Molecules.